# Efficiency of Water Pollution Control Based on a Three-Stage SBM-DEA Model

Yongdi Chen [1], Chunhui Li [2], Xiaoxuan Li [2], Xiaolan Zhang [3] and Qian Tan [1,*]

[1] Key Laboratory for City Cluster Environmental Safety and Green Development of the Ministry of Education, School of Ecology, Environment and Resources, Guangdong University of Technology, Guangzhou 510006, China; 15626138403@163.com

[2] School of Environment, Beijing Normal University, Beijing 100875, China; chunhuili@bnu.edu.cn (C.L.); lxxxlbnu@163.com (X.L.)

[3] Foreign Environmental Cooperation Center, Ministry of Ecology and Environment, Beijing 100035, China; zhang.xiaolan@fecomee.org.cn

[*] Correspondence: qian_tan@gdut.edu.cn

**Abstract:** With the growing severity of water pollution issues, the prevention and control of water pollution became highly complicated and challenging, and the investment in water pollution control has been constantly increased. Scientific evaluation of efficiency is critical to recognize whether the investments in water pollution control are effective. However, most studies could not exclude the influences of external environmental and random factors when evaluating the efficiency of water pollution control, resulting in biased results. To overcome this shortcoming, this study employed a three-stage SBM-DEA (slacks-based measure-data envelopment analysis) model to determine the efficiency of water pollution control efforts in a city of China from 2003 to 2017. The results showed that water quality in the study area has been significantly improved due to those pollution control efforts. The influences from external environmental and stochastic factors have led to an underestimation of the efficiency of water pollution control in the first stage. After excluding these effects in the second stage, the adjusted efficiency of water pollution control showed a fluctuating upward trend in the third stage, reflecting the true effectiveness of efforts to prevent and control water pollution in the study cities, with an average efficiency of 0.87. Finally, several suggestions for enhancing the efficiency of water pollution control in Chengde were proposed.

**Keywords:** water pollution control efficiency; three-stage SBM-DEA model; external environment factors; data envelopment analysis

## 1. Introduction

Water pollution has long been one of the major concerns threatening environmental security [1,2]. Water pollution has also become a prominent issue worldwide that restricts the socio-economic development of watersheds and regions and compromises public health [3,4]. With the rapid development of urbanization, industrialization, and agricultural modernization, prevention and control of water pollution has become highly complicated and challenging [5]. As a result, investments in technologies and facilities in water pollution control have been continuously increased [6]. It is of significance to assess if and how these investments in water pollution control are effective.

The efficiency of pollution control is considered as a critical index that reflects the effectiveness of pollution prevention and control efforts [7], and it was defined as the relative effectiveness between the input and output elements in the prevention and control of environmental pollution [8]. Studies on the efficiency of pollution control have already received considerable academic attention, especially in the measurement of pollution control efficiency [9]. In terms of methods, the main evaluation methods applied to measure the efficiency of pollution control included stochastic frontier analysis (SFA) and data

envelopment analysis (DEA). The SFA method is a representative of the parametric model, which is one of the essential methods to deal with multi-input single-output models. For example, Zheng et al. investigated the investment efficiency of environmental governance in China through the SFA model and explored its spatial and temporal evolution [10]. Zhu et al. used the SFA method to measure the environmental efficiency of 66 cities in eastern China and investigated the impact of collaborative agglomeration of the industry on efficiency [11]. The SFA method is capable of minimizing the influence of random factors over the results due to its stochastic frontier. Nevertheless, the SFA model requires the pre-setting of production functions and various parameters before evaluation, and it is hard to ensure that these settings are in accordance with real-world problems [12]. Besides, the SFA model has difficulty in solving cases with multiple inputs and outputs. In contrast, the DEA method can address these drawbacks well, as DEA is a non-parametric model that evaluates the relative effectiveness of the production possibility set consisting of decision units through mathematical programming, which eliminates the need to set production functions and estimate parameters [13]. This model enables efficiency assessment with multiple inputs and outputs and objective weighting based on available data, and is thus the most commonly employed method for evaluating the efficiency of pollution control [14].

With the development of DEA methods, the application of different DEA models in the evaluation of pollution control efficiency has continuously been advanced. For instance, Simoes et al. used a radial DEA model to study the operational performance of 29 solid waste utilities in Portuguese and found that environmental contexts such as gross domestic product (GDP) per capita had a significant effect on the operational performance [15]. Mandal et al. also used a radial DEA model in combination with a directional distance function to evaluate the environmental management efficiency of rural industrial pollution [16]. The radial DEA model ignores the slack variables and unreasonably assumes proportional change in inputs and outputs, which may lead to biased measurement results. Given this, Tone proposed a slacks-based measure DEA (SBM-DEA) model, which has become one of the mainstream methods for evaluating efficiency in various fields, including the efficiency of pollution control [17]. Cheng et al. conducted an efficiency evaluation of 681 wastewater treatment facilities in China through the SBM-DEA model, and found that only 27 wastewater treatment facilities in China reached the DEA effective frontier [18]. Yang et al. measured the wastewater control efficiency of 39 industrial sectors in China with a SBM-DEA model and obtained more accurate results for the total factor efficiency through a multi-constrained nonlinear function [19]. Wu et al. combined the SBM-DEA model and an improved Luenberger productivity decomposition method, determining the wastewater control efficiency at the urban scale in China from both static and dynamic aspects, respectively [20]. In addition, there were some novel DEA models such as the network DEA model and uncertainty DEA model gradually applied to the efficiency evaluation of water or air pollution control [21,22]. As a non-radial and additive DEA method, the SBM model provides a more accurate measurement for pollution control efficiency. Nevertheless, this model is incapable of removing the influences of external environmental factors and random errors from the evaluation process, which may lead to biased results.

It has been confirmed by many studies that there were multiple external environmental factors showing a significant impact on the efficiency of pollution control. Those factors mainly included urbanization level [23], population growth [24], economic development level [25], industrial structure [26], environmental regulation [27], technological progress [28], public participation [29], and others. To address the above deficiencies, Fried et al. combined DEA and SFA methods to develop a three-stage DEA model that considers the impact of environmental factors and random noise on the efficiency evaluation of decision units [30]. The model uses a traditional DEA model to obtain the slack information of indicators in the first stage. Based on this, the SFA model is adopted in the second stage to identify the effects of environmental factors and random noise to adjust the indicators, and in the third stage, adjusted efficiency is measured by these modified indicators. Some

researchers have made attempts to study the pollution control efficiency by using the three-stage DEA method. For instance, Gong et al. measured the environmental and economic efficiency of air pollution control in 30 Chinese provinces by means of a three-stage DEA model, and the results showed that the economic efficiency of air pollution control was more reliable after removing the interference [23]. Peng et al. used the three-stage super-SBM model to investigate the efficiency of urban environmental governance and explored the regional difference through the calculation of the DagumGiniratio index [31]. Shi used a three-stage DEA model to measure the efficiency of industrial water pollution control in 28 provinces of China, and found that the interference from environmental variables and stochastic factors can lead to an overestimate of efficiency [32]. However, studies on the efficiency of water pollution control by using three-stage DEA method are still scarce, especially at the regional scale. This is because previous studies mostly focused on a certain type of water pollution control rather than all the water pollution control efforts in a region.

To sum up, although much effort has been made in the evaluation of pollution control efficiency, up-to-date research still suffers from the following deficiencies. Firstly, most studies mainly adopted the traditional DEA method to evaluate the efficiency of water pollution control, ignoring the impact of external environmental and random factors in the evaluation process, which probably leads to biased results. Secondly, previous studies tended to choose a certain type of pollution as a decision unit, making it difficult to sufficiently represent all efforts of water pollution control at a regional scale.

Therefore, this study considered the total inputs and outputs of water pollution control efforts at a regional scale, and introduced a three-stage SBM-DEA model that can eliminate the influence of external environmental and stochastic factors over the efficiency of water pollution control. The application to Chengde city in China revealed the effectiveness of local water pollution control activities from 2003 to 2017. In addition, through the analysis of the influencing factors of water pollution control efficiency, recommendations to achieve efficiency improvement in Chengde city are proposed.

## 2. Methodology

### 2.1. Three-Stage SBM-DEA Model

In this study, we employed a three-stage SBM-DEA approach to measure the efficiency of water pollution control in Chengde city. By referring to the study of Zhang et al. [33], the analytical framework based on the three-stage SBM-DEA model constructed for this study is shown in Figure 1.

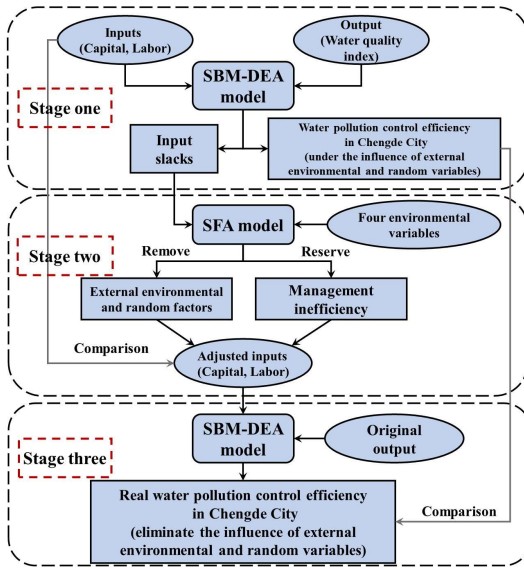

**Figure 1.** The analytical framework.

### 2.1.1. Conventional SBM-DEA Model

In the first stage, the conventional slack-based measure (SBM) DEA model was adopted in this study to measure the efficiency of water pollution control in Chengde city. Data envelopment analysis (DEA) is a nonparametric statistical method for evaluating relative efficiency proposed by American operations researchers Charnes et al. in 1978, mainly for decision units with the same type of multiple inputs and outputs [34]. These traditional radial DEA models assumed the same proportional variation of inputs and outputs, which is not consistent with the actual production process. In order to avoid the neglect of slack variables and the influence caused by the radial DEA model, Tone et al. first proposed a non-radial SBM-DEA model in 2001 [17]. This model has become the mainstream model for efficiency evaluation in various fields, and its specific expression is shown in model (1)

$$
\begin{aligned}
min\rho &= \frac{1 - \frac{1}{m}\sum\limits_{i=1}^{m} s_i^- / x_{i0}}{1 + \frac{1}{s}\sum\limits_{j=1}^{s} s_j^+ / y_{j0}} \\
\text{s.t.} x_0 &= X\lambda + s^- \\
y_0 &= Y\lambda - s^+ \\
\lambda &\geqslant 0, s^- \geqslant 0, s^+ \geqslant 0
\end{aligned}
\tag{1}
$$

where $\rho$ represents the efficiency value. $m$ and $s$ respectively denote the number of input and output indicators of the decision unit. $s^-$ and $s^+$ denote slack variables for input and output indicators, respectively. $x_0$ and $y_0$ indicate the input and output vectors of the decision unit, respectively. $X$ and $Y$ are the input and output matrices composed of all decision units, respectively. Only when the slack variables $s^- = 0, s^+ = 0$, can the decision unit be called efficient; otherwise, it is inefficient, implying that it has potential for improvement.

### 2.1.2. SFA Model for Inputs Adjustment

As the efficiency evaluation of the decision unit in the first stage was influenced by the combination of environmental factors, management factors, and random errors, the evaluation results might be biased [30]. In order to further distinguish the degree of influence of each factor, the SFA model was required in this stage to eliminate the environmental factors and random error to improve the reliability of the DEA evaluation. Then, the input redundancy of decision unit was only caused by the management inefficiency. Assuming that there are $n$ decision units, each with $p$ inputs and $m$ observable external environmental influence factors, the SFA regression equation can be constructed as follows:

$$
S_{ik} = f_i(z_k; \beta_i) + v_{ik} + \mu_{ik} (i = 1, 2, \cdots, m; k = 1, 2, \cdots, n)
\tag{2}
$$

where $S_{ik}$ denotes the slack variable of input $i$ for decision unit $k$. $f_i(z_k; \beta_i)$ is the effect of environmental variables on the slack variable $S_{ik}$ of input $i$, with the general form of $f_i(z_k; \beta_i) = z_k\beta_i$. $z_k = \left(z_{1k}, \ldots, z_{pk}\right)$ are the observable environmental variables of decision unit $k$. $\beta_i$ is the coefficient to be estimated for the environmental variables. The joint term $v_{ik} + \mu_{ik}$ is the mixed error, where $v_{ik}$ denotes the random error with a distribution of $v_{ik} \sim N\left(0, \sigma_{vi}^2\right)$. $\mu_{ik}$ is the management inefficiency term with a distribution of $\mu_{ik} \sim N^+\left(\mu_i, \sigma_{\mu i}^2\right)$. $v_{ik}$ and $\mu_{ik}$ are independent of each other and uncorrelated. Assuming $\gamma = \sigma_{\mu i}^2 / \left(\sigma_{\mu i}^2 + \sigma_{vi}^2\right)$, this term is the proportion of the variance of management inefficiency to the total variance. When the value of $\gamma$ tends to 1, it indicates that the management inefficiency factor is the main influence. When the value of $\gamma$ tends to 0, it indicates that the influence of random error factor is more significant. At this point, $\mu_{ik}$ can be eliminated and the stochastic model can be changed into a deterministic model, which can be estimated by the ordinary least squares (OLS) model. In order to separate the random error from the management

inefficiency in the mixed error term, the maximum likelihood estimation is first performed using the software Frontier 4.1 to obtain an estimate of $\beta_i, \gamma, \sigma^2$.

Then, an estimate of the management inefficiency is derived according to the method proposed by Jondrowet et al. [35]

$$\hat{E}[\mu_{ik} \mid v_{ik} + \mu_{ik}] = \frac{\gamma\sigma}{1 + \gamma^2} \left( \frac{\varphi(\gamma e_i)}{\phi(\gamma e_i)} + \gamma e_i \right) \tag{3}$$

where $\phi$ and $\varphi$ are the distribution function and density function of the standard normal distribution, respectively. $e_i$ is the error term, and the estimate of $v_{ik}$ can be further obtained.

$$\hat{E}[v_{ik} \mid v_{ik} + \mu_{ik}] = S_{ik} - f_i(z_k; \hat{\beta}_i) - \hat{E}[\mu_{ik} \mid v_{ik} + \mu_{ik}] \tag{4}$$

According to the results of SFA model, the slack variables of the inputs are determined based on the degree of influence of each environmental variable. Based on this, the corresponding input adjustments are applied to the decision units that do not reach the effective frontier

$$x_{ik}^* = x_{ik} + \left[ max_k\{z_k\hat{\beta}^n\} - z_k\hat{\beta}^n \right] + \left[ max_k\{\hat{v}_{ik}\} - \hat{v}_{ik} \right] \tag{5}$$

where $x_{ik}^*$ and $x_{ik}$ are the adjusted and original values of input $i$ for decision unit $k$, respectively, and $\hat{\beta}^n$ denotes the estimated value of the environmental variable. The second term on the right side of Equation (5) represents the increment of the input $i$ of all decision units in the case where the environmental variables act as the maximum, i.e., in the worst environment. The third term on the right side of this equation is the increment of the input placed in the case of the maximum random errors. Thus, each decision unit can be guaranteed to face the same external conditions.

### 2.1.3. Improved SBM-DEA Model for Efficiency Measurement

In this stage, this study replaced the original input $x_{ik}$ with adjusted input $x_{ik}^*$ for removal of environmental and stochastic factors in the second stage. The rest of the calculation steps were maintained as in stage 1, continuing to adopt model (1) to determine the efficiency of water pollution control in Chengde city. By using the information contained in the slack variables, the efficiency results measured in this stage excluded the influence of external environmental and stochastic factors, and were the actual performance of the efficiency of water pollution control in Chengde city over the years.

### 2.2. Input and Output Indicators

Based on the principles of systematicity, comprehensiveness, and homogeneity of indicators selected for efficiency evaluation and the availability of data related to water pollution prevention and control in Chengde, the specific indicators selected for measuring the efficiency of water pollution control in this study were as follows.

**Input indicators.** The input indicators were composed of capital input and labor input. Capital input, a direct reflection of the amount of capital investment, was indicated by the regional total investment in water pollution prevention and control. The total investment should be able to fully cover all aspects of water pollution control efforts. The total investment in water pollution prevention and control in Chengde city mainly consisted of the investment in water conservancy projects and the partial investment in ecological environmental protection. Labor input was represented by the number of employees related to water pollution prevention and control in Chengde city. The labor input was mainly composed of employees of water conservancy project management and, in part, of employees of ecological environmental protection business. Based on the data availability and references to Sun et al. [36], the specific calculation of capital and labor input indicators was as follows:

$$x_1 = a_1 + \eta b_1$$
$$x_2 = a_2 + \eta b_2 \tag{6}$$

where $x_1$ and $x_2$ denote indicators of capital and labor input, respectively. $a_1$ and $a_2$ indicate the amount of investment and the number of employees for the water conservancy project management in Chengde city, respectively. $b_1$ and $b_2$ denote the total amount of investment and the number of employees for ecological and environmental management business in Chengde city, respectively. $\eta$ is the proportion coefficient of water pollution control business to environmental and ecological protection business in Chengde city. Based on the advice of the staff of the Chengde Ecological Environment Bureau, the value is taken as: $\eta = 0.25$.

**Output indicator**. The output indicator was indicated by the comprehensive water quality index. The fundamental purpose of water pollution prevention and control is to improve the water quality. That is to say, it was considerably appropriate and scientific for this study to select the comprehensive water quality index as the output indicator. Referring to Yuan et al. [37], this study adopted the scoring method to quantify the water quality condition of surface water in Chengde city, and then obtained the comprehensive water quality index, which was calculated by the following formula:

$$y = \sum c_i \cdot m_i / \sum m_i \tag{7}$$

where $y$ represents the comprehensive water quality index. $m_i$ indicates the number of monitoring sections of different water qualities. $c_i$ denotes the score of different water qualities, class I to inferior V water quality score in the order of 5 to 0 points.

### 2.3. Environmental Impact Factors

Different empirical analyses have been conducted by researchers on the influencing factors of pollution control efficiency. In most of these previous studies, the economic development level and industrial structure were the explanatory variables explored the most frequently for efficiency of pollution control [33,38–40]. In addition, as local policy makers and regulators, the government's influence and intervention are crucial to pollution control, which has received attention in many studies [23,32]. Therefore, based on the existing research, this study was designed to investigate the possible effects of four external environmental variables in the following three categories on efficiency of water pollution control.

**Economic development level**. According to the environmental Kuznets curve, it is known that the environment and the economic development level have an inverted U-shaped relationship. That is, with the increase of gross domestic product (GDP) per capita, people's demand for improving environmental quality is more urgent, and pollution control will be attached more importance. In addition, a high GDP per capita generally means that there is sufficient fiscal revenue to invest in environmental protection, facilitating the improvement of pollution control. Here, GDP per capita was chosen to represent the environmental impact variable of economic development level.

**Industrial structure**. Generally speaking, the accelerated industrialization process emits many pollutants, which will bring influence and pressure to the local pollution control, while the development of tertiary industry is helpful to alleviate the pressure of environmental pollution. Therefore, this study selected the proportion of secondary industry (calculated from the share of industrial value added in GDP) and the proportion of tertiary industry (calculated from the share of total tertiary output in GDP) to investigate the influence of industrial structure on the efficiency of water pollution control.

**Government influence**. The government plays an essential leading role in environmental management. For one, the government can curb the emission of pollutants from enterprises and improve environmental quality by enacting regulations. For another, the environmental regulations set by the government may impose additional economic costs.

In this study, the fiscal decentralization degree (calculated from the share of local fiscal expenditure in GDP) was chosen to represent the government influence.

### 3. Study Problem

Chengde city is located in the northeastern part of Hebei province in China, with a total land area of 39,500 km², accounting for one-fifth of the total land area of Hebei province. There are four major river systems in Chengde, including Luanhe River, Chaohe River, Liaohe River, and Dalinghe River, with an annual water production of about 3.76 billion m³. The main reasons for selecting Chengde as the study area in this paper are as follows. Firstly, with its special geographical location, Chengde is an important source of water supply for major cities in the Beijing-Tianjin region of China. As a significant water source and water conservation area, the water quality in Chengde directly affects the water security of Beijing-Tianjin region. Secondly, Chengde city is of special strategic significance and belongs to one of the pilot areas of ecological civilization construction in China. The pilot construction experience of Chengde is very valuable for exploring the development path of ecological civilization in northern China, and can play a typical demonstration role for the ecological civilization construction in China. In addition, Chengde city has invested a huge amount in the prevention and control of water pollution, while the effectiveness of water pollution has not been scientifically identified. There were more than 250 ecological engineering projects conducted in Chengde in the past decade, with a total investment of nearly 30 billion yuan. Thus, it is urgent and essential to evaluate the efficiency of water pollution prevention and control in Chengde over the years.

The study area is shown in Figure 2.

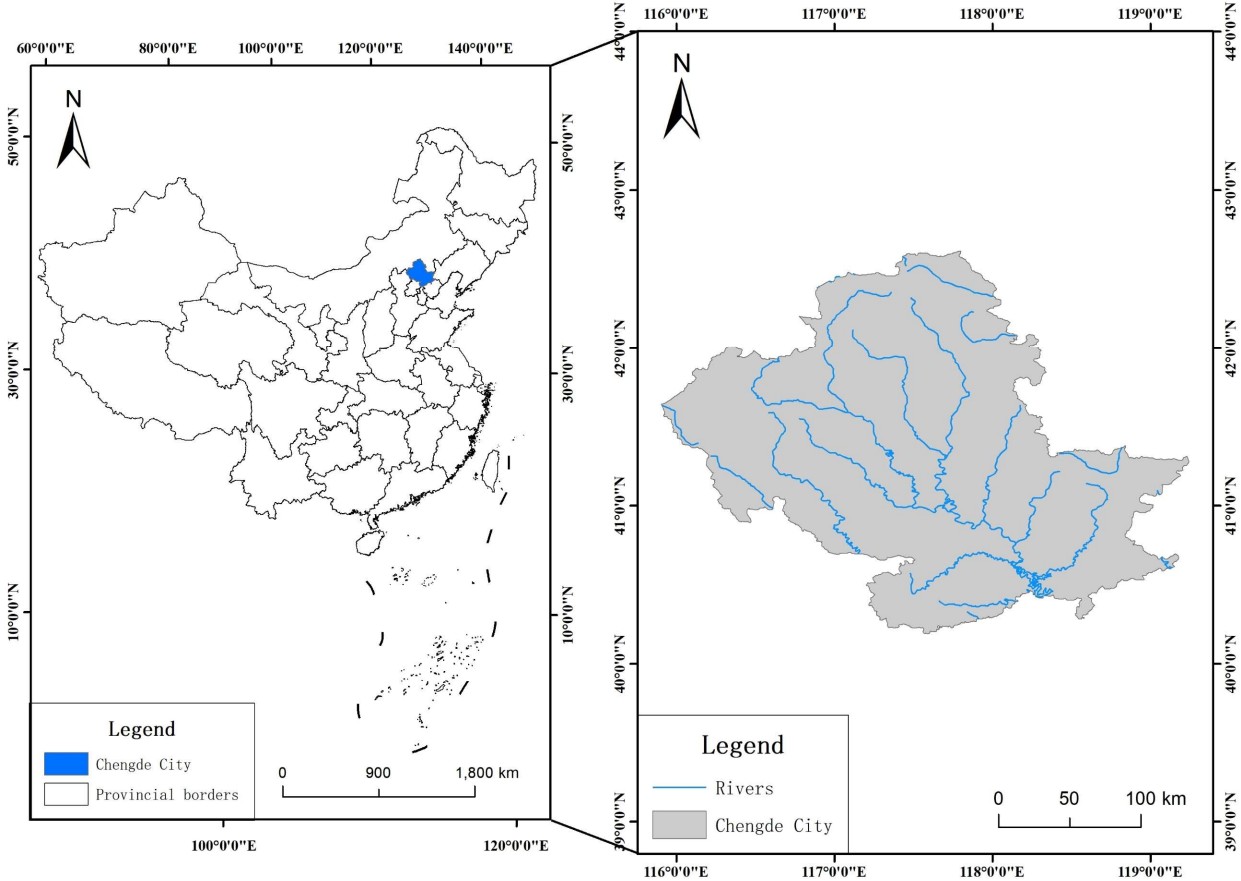

**Figure 2.** The study area.

The data of environmental impact variables and input and output indicators mentioned above were obtained from the Chengde Statistical Yearbook, Hebei Statistical Yearbook and relevant information provided by the Chengde Ecological Environment Bureau in previous years. The descriptive statistical results of each variable are shown in Table 1.

**Table 1.** Results of descriptive statistics for each variable.

| Variable Type | Indicator | Mean | Standard Errors | Minimum | Maximum |
|---|---|---|---|---|---|
| Input | Capital input/million yuan | 2068.51 | 2164.44 | 166.08 | 6768.98 |
| | Labor input/person | 1647.10 | 389.22 | 1205.50 | 2247.75 |
| Output | Comprehensive water quality index | 2.41 | 0.71 | 1.04 | 3.48 |
| Environmental variable | GDP per capita/yuan | 25,623.07 | 12,117.37 | 6191.00 | 41,299.00 |
| | Secondary industry share/% | 44.27 | 4.49 | 33.88 | 52.84 |
| | Tertiary sector share/% | 33.75 | 3.19 | 30.18 | 42.33 |
| | Fiscal decentralization degree/% | 17.26 | 3.37 | 13.20 | 23.04 |

## 4. Results and Discussion

### 4.1. Changes in Water Quality

As the output indicator reflecting the surface water quality condition of Chengde city, the comprehensive water quality index can be obtained from the calculation of water quality scoring and the number of sections of water quality monitoring. Based on the obtained data, the comprehensive water quality index of Chengde city from 2003 to 2017 could be determined by Equation (7). As shown in Figure 3, the comprehensive water quality index of Chengde city showed a fluctuating upward trend during the study period, indicating that the water environment quality of Chengde city was improving over time. Specifically, the percentage of monitoring sections with water quality of III categories or better in Chengde improved from 0% in 2003 to 79% in 2014. In each five-year planning period, there was a significant enhancement in the water quality index. Especially in the last few years (from 2014 to 2017), the water environment quality in Chengde has stabilized at a high level with the water quality index greater than 3. This is because the prevention and control of water pollution in Chengde has achieved breakthrough success, i.e., the monitoring sections with water quality of V categories or worse was cleared in 2014.

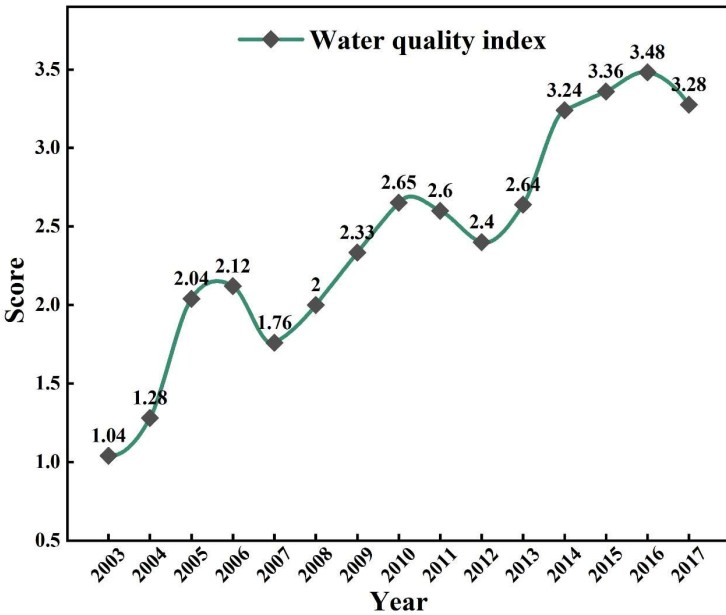

**Figure 3.** Changes of water quality in Chengde from 2003 to 2017.

As a matter of fact, since the Tenth Five-Year Plan, Chengde city has been vigorously promoting the comprehensive management of water environment in order to improve water quality. In the early period of water pollution prevention and control, Chengde mainly controlled the point source pollution by constructing wastewater treatment plants and improving sewage pipelines. The water quality improvement of the monitoring section downstream of the sewage treatment plant demonstrated the effectiveness of such efforts. Further, in 2012, Chengde city implemented a series of planning including the "Program for the Prevention and Control of Water Pollution in Chengde City" and "Planning for the Prevention and Control of Water Pollution in Luan River Basin", which target non-point sources of pollution such as soil erosion and livestock breeding. In terms of the agriculture, Chengde city implemented zero-growth action of pesticides and fertilizers, and promoted a new model of irrigation and fertilization. Through the control of point source and non-point source pollution, and the combination of engineering measures and various programs and policies, the surface water quality in Chengde has been further improved. In general, judging from the results of the comprehensive water quality index, it is worthwhile to confirm the efforts of water pollution prevention and control in Chengde city, which played an exemplary role in the experience and modeling of water pollution control to some extent.

### 4.2. Water Pollution Control Efficiency without Input Indicators Adjustment

In this stage, based on the acquired data and through the software DEA-solver 13.0, the SBM model was adopted to determine the efficiency of water pollution control in Chengde city between 2003 and 2017, with the results shown in Figure 4. In addition, this paper also measured the slack variables of input and output indicators for the purpose of explaining the inefficiency sources, with the results shown in Figure 5.

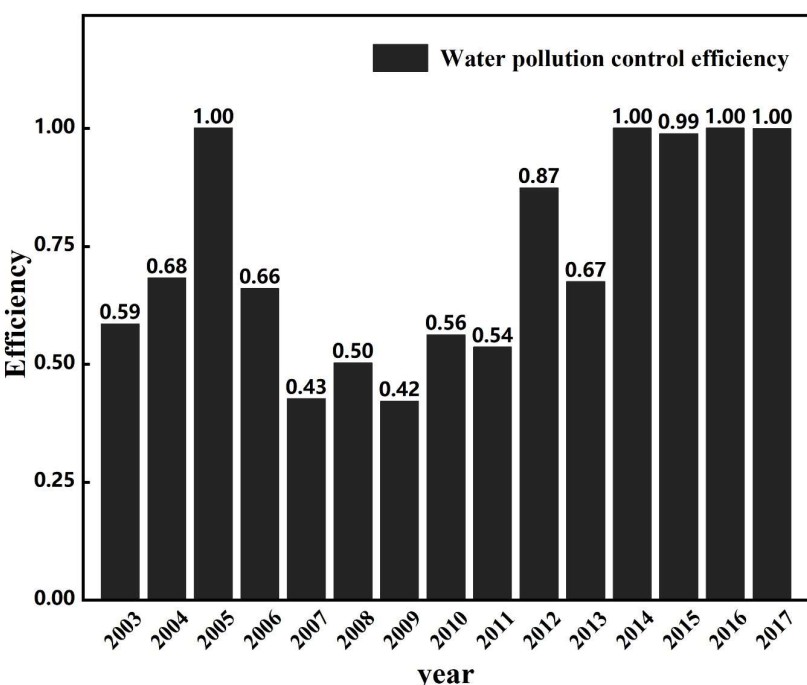

**Figure 4.** Water pollution control efficiency in Chengde city from 2003 to 2017.

As illustrated in Figure 4, without excluding external environmental variables and stochastic factors, there were four years (including 2005, 2014, 2016, and 2017) in Chengde city that reached the technically effective frontier. That is, the input-output ratio achieved the optimal state with its efficiency equal to 1. The efficiency of the remaining years was non-DEA effective, and the lowest efficiency of water pollution control was in 2009, which was only 0.42. In terms of changes, the water pollution control efficiency of Chengde city exhibited a general fluctuating upward trend between 2003 and 2017. In 2005, the efficiency

of water pollution control quickly rose to 1, reaching DEA effectiveness. Then, the efficiency dropped to the lowest point in 2009 and began to fluctuate and grow over time, finally stabilizing at the high efficiency level, which is similar to the change of water quality index. However, this result did not eliminate the influence of environmental and random factors, and failed to reveal the actual status of water pollution control efficiency in different years in Chengde, so further adjustment and measurement were necessary.

As presented in Figure 5, the capital investment in water pollution control in Chengde was excessive in most years, indicating that the redundancy of capital inputs is an important reason for the inefficiency of water pollution control. Especially in 2009 and 2013, the excess of capital input was the largest, with more than half of the inputs not performing effectively, which could account for the sharp efficiency decline in these two years. Compared to the former year, the increase in capital investment was 420% and 230% in 2009 and 2013, respectively. The excessive increase in capital investment may have led to an irrational use of funds. In terms of the labor input, only in 2003 and 2006 there was redundant labor input, suggesting the redundancy of labor input is not the critical reason for the inefficiency of water pollution control. In other words, Chengde's manpower inputs in water pollution control are effective. However, in order to ensure that the efficiency of water pollution control can steadily achieve DEA effectiveness, it is essential for Chengde City to emphasize the enhancement of the competence of employees in water pollution control while focusing on the management of the number of employees.

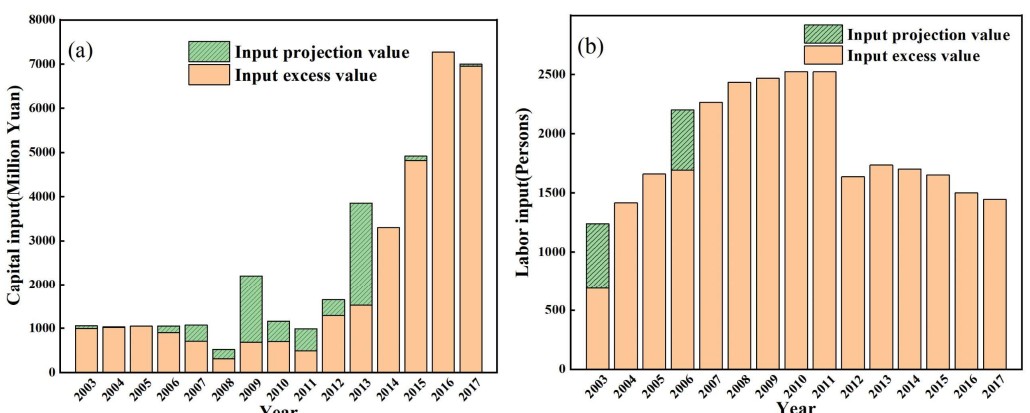

**Figure 5.** Projection and excess value of capital (**a**) and labor (**b**) inputs of water pollution control.

As shown in Figure 6, there were six years of shortfalls in output for water pollution control in Chengde, including 2004, and from 2007 to 2011. This suggested that the water quality improvement efforts in these years did not achieve fully satisfactory results. The inefficiency in 2009 was also due to the inability to improve water quality significantly. This is because the actions to build and upgrade wastewater treatment plants showed limited improvement in water quality, which might prove that the key issue of water pollution control in 2012 was not the source of pollution anymore. In summary, the excess of input indicators and the shortfall of output indicators together resulted in the efficiency of water pollution control not reaching the effective production frontier. It is extremely crucial for Chengde to allocate funds reasonably in water pollution prevention and control, as well as to further improve and optimize the management of technologies in water pollution prevention and control, which is conducive to the improvement of water pollution control efficiency.

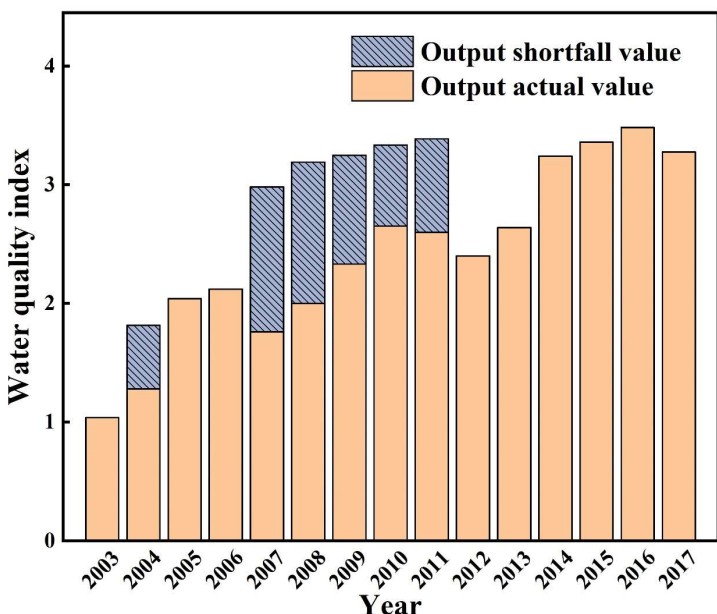

**Figure 6.** Shortfall and actual value of output of water pollution control from 2003 to 2017.

### 4.3. Analysis of SFA Regression Results

In the second stage of DEA analysis, by using the SFA model to decompose the impact of environmental factors, random errors, and management inefficiencies on water pollution control efficiency, the original input value over the years could be adjusted, and finally we could obtain the actual level of water pollution control efficiency under the same management environment. The slack of the capital input and labor input measured in the first stage was taken as the explained variable in the regression function, and the per capita GDP, the proportion of the secondary industry, the proportion of the tertiary industry, and the fiscal decentralization degree were used as the explanatory variables. This study explored the influence of four variables on the slack variables of two input indicators by using the software Frontier4.1, and the regression results of SFA model in the second stage could be obtained, as shown in Table 2. It can be seen from Table 2 that all the results were significant at the 1% confidence level. Thus, the external environment factors all had a significant impact on the slack variable of inputs in the water pollution prevention and control in Chengde. Therefore, in order to effectively strip management factors and random factors, it was important to adjust the input variables in the second stage.

**Table 2.** The regression results of SFA model in the second stage.

| Variables | Coefficient of Capital Input | Coefficient of Labor Input |
|---|---|---|
| Constant terms | −582,105.53 *** (−582,105.53) | −759.32 *** (147.20) |
| GDP per capita | 808,273.1 *** (808,273.1) | 223.82 *** (95.11) |
| The secondary industry share | 432,829.71 *** (432,829.71) | 17.02 *** (8.73) |
| The tertiary industry share | 1.76 *** (51.89) | 0.01 *** (12.97) |
| Fiscal decentralization degree | 6000.54 *** (6000.54) | 325.76 *** (33.55) |
| $\sigma^2$ | $8.63 \times 10^9$ *** ($8.63 \times 10^9$) | 32,519.73 *** (32,517.58) |
| $\gamma$ | 1.00 *** (1,619,404.5) | 1.00 *** (134.23) |
| Log likelihood | −181.92 | −89.17 |
| LR test | 8.97 | 8.98 |

Note: data in parentheses indicate t-statistics. "*", "**", and "***" indicate significant levels at 10%, 5%, and 1%, respectively.

When examining the impact of environmental variables on the input slack variable, if the coefficient result is positive, it means that the increase in the value of the environmental variable will lead to an increase in the input slack variable or a decrease in output, resulting

in an increase in waste and adversely affecting the efficiency of water pollution control. If the coefficient result is negative, it indicates that the rise of this environmental variable will bring about a reduction in the input slack variable or an increase in output, resulting in savings and a positive impact on the efficiency of water pollution prevention and control. The following analyzes the effect of each specific significant environmental variable on the input slack variable.

**GDP per capita.** Table 2 showed that the regression results of per capita GDP on the slack variables of labor and capital inputs were all positive, and the regression coefficients of these two input slack variables were both at the 1% significance level. This meant that the increase in per capita GDP would trigger an increase in the slack variables of labor and capital inputs, thereby decreasing the efficiency of capital and labor utilization in water pollution prevention and control. This result was not consistent with the environmental Kuznets curve and was contrary to the findings of Gong et al. [23]. This might be mainly due to the threshold effect of the increase in per capita GDP on the efficiency of water pollution control. When it is lower than the threshold value, the increase of per capita GDP would not have a positive impact on the improvement of capital and labor utilization efficiency, which would not promote the efficiency of water pollution control. In addition, the increase in GDP per capita might lead to the irrational use of capital, resulting in input redundancy, which makes the efficiency of water pollution control decrease.

**Secondary industry.** The calculation results indicated that the regression coefficients of the proportion of the secondary industry on the slack variables of labor and capital inputs were all positive values, and they were all significant at the 1% level. It could be seen that the increase in the proportion of the secondary industry would cause an increase in the slack variables of labor input and capital input, resulting in a decrease in the efficiency of water pollution control, which is consistent with the findings of Zhang et al. [33]. The greater the proportion of the secondary industry, the more the various pollutants produced by industrial production, and the more significant the harm to the water environment. To a certain extent, this might increase the additional marginal costs of water pollution control, leading to a decrease in capital utilization and thus a reduction in the efficiency of water pollution control, which is consistent with the actual situation. At the same significance level, the regression coefficient of the proportion of the secondary industry on the slack variable of capital investment was much larger than that of the labor input, which well showed that the utilization efficiency of capital was affected by the proportion of the secondary industry more significantly. The possible reason for this might be the low innovation ability of the treatment technology for industrial water pollution in Chengde, which leads to the poor efficiency of capital utilization.

**Tertiary industry.** The proportion of the tertiary industry had a significant positive effect on the labor and capital inputs slack variables at a significance level of 1%, which indicated that the increase in the proportion of the tertiary industry would bring an increase in the slack variables of labor input and capital inputs. The regression coefficients of the proportion of the tertiary industry on the two input slack variables were both small, where the coefficient of labor input was only 0.01, which was much smaller than the coefficient of capital input. Generally, the increase in the proportion of the tertiary industry facilitates the reduction of pollution caused by agricultural and industrial production, and is conducive to improving the efficiency of water pollution control. However, the current results were contrary to this traditional cognition. The reason might be that Chengde, as a tourist city, was dominated by the development of tertiary industries, and its tertiary industry accounted for 42.3% in 2017, which is larger than the 41.7% of the secondary industry. Compared with other typical cities, the scale of tertiary industries such as tourism and catering were larger in Chengde, which has caused a larger pollution burden to the water environment to a certain extent. In addition, the loosened regulation of the tertiary industry might also cause accidental water pollution and led to a decrease in the efficiency of water pollution control.

**Financial decentralization degree.** The government influence indicator was represented by the degree of fiscal decentralization. The regression results illustrated that at the 1% significance level, the regression coefficients of government influence on labor and capital inputs slack variables were both positive. A high degree of fiscal decentralization would promote the increase of the slack variables of these two input indicators, which was not conducive to the improvement of the efficiency of water pollution prevention. This might be caused by the deficiencies of the administrative management system. The higher the degree of fiscal decentralization, the higher the freedom of the government to manage financial resources, which may lead to the unreasonable allocation of fiscal funds in the basic public services with strong positive externality such as environmental governance, ultimately decreasing the capital use efficiency in water pollution control.

By comparing the absolute values of the significant coefficients of different external environmental variables for the same input index, the order of importance of each environmental variable for moderation could be obtained. For capital investment, the secondary industry had the highest priority, followed by per capita GDP, fiscal decentralization, and the tertiary industry. For labor input, the priority of fiscal decentralization was the highest, followed by GDP per capita, the proportion of the secondary industry, and the proportion of the tertiary industry. Based on the above analysis, it is easy to know that environmental variables have significant effects on different input slack variables and the estimated water pollution prevention and control efficiency will be biased due to the effect of external environmental factors. Therefore, it is necessary to adjust the initial input variables to remove the effects of environmental and random factors, and then explore the actual level of water pollution control efficiency.

*4.4. Water Pollution Control Efficiency with Input Adjustment*

In this stage, two input variables were corrected according to Equation (4), and the adjustment results are shown in Figure 7. By using the software DEA-solver 13.0, the adjusted input variables and the original output were measured again with the SBM-DEA model. The actual efficiency of water pollution control in Chengde city for all years was obtained, and the specific results are displayed in Figure 8.

As shown in Figure 7, the adjusted capital and labor inputs were basically greater than the original values in each year, which verified the influence of external environmental variables and random errors on the evaluation of water pollution control efficiency. For the capital input, both the original and adjusted values showed an increasing trend of fluctuation over time, which indicated that the water pollution control in Chengde was getting more and more attention. By comparing the difference between the original and adjusted values, it was found that the adjustment of capital input was relatively large from 2003 to 2006, indicating that the capital input in these years was significantly influenced by the combination of external environmental variables and random errors. For the labor input, both the original and adjusted values showed a trend of increasing and then decreasing. This might demonstrate the transition of labor input in Chengde from a manned workforce to a small but efficient workforce. The difference between the original and adjusted values showed that the adjustment of labor input was relatively large from 2005 to 2011, which was caused by the significant influence of environmental factors and random errors on the slack variables of labor input.

As shown in Figure 8, by comparing the results of stage 1 with stage 3, it can be observed that after removing the combined effect of external environment and stochastic factors, the efficiency of water pollution control in some years in Chengde city showed a large change. The adjusted efficiency of water pollution control in some years (from 2006 to 2013) was significantly higher than the initial efficiency value, indicating that the efficiency in these years was clearly underestimated in the first stage, which implies that it is of great necessity to exclude the effects of external environment and stochastic factors performed in the second stage. There were three years (including 2004, 2015, and 2017) where the adjusted efficiency of water pollution control was slightly lower than the original

efficiency value, indicating that these efficiencies were slightly overestimated in stage 1. In terms of the number of decision units whose efficiency reached DEA effectiveness, the number of effective decision units in the third stage was three, less than the four in the first stage. The year 2017, which was considered DEA effective in the first stage, did not achieve DEA effectiveness after the adjusted measurement in the third stage. Similarly, the efficiency of water pollution control measured in the third stage and the first stage both exhibited an overall fluctuating upward trend during the study period, and reached a high efficiency status in the recent years. Differently, the mean value of efficiency measured in the third stage was 0.87, greater than 0.72 in the first stage, and the efficiency variation among decision units decreased compared with the first stage, which suggested an actual excellent level of water pollution prevention and control in Chengde.

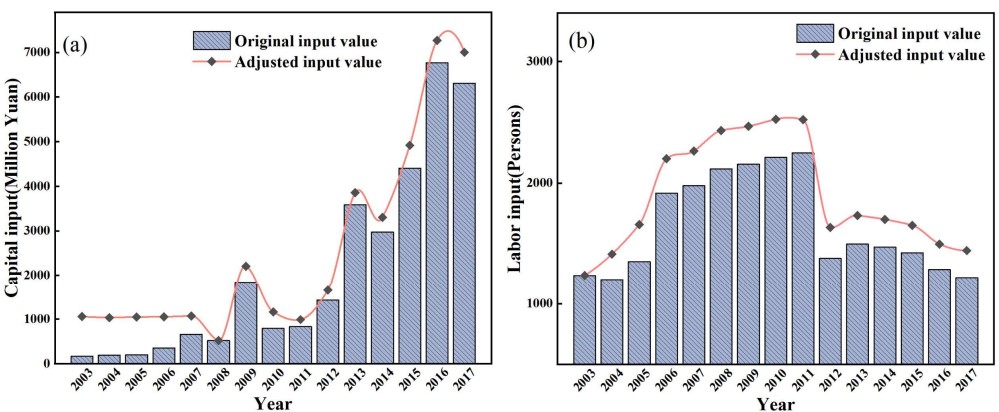

**Figure 7.** Comparison of the original and adjusted capital (**a**) and labor (**b**) inputs of water pollution control.

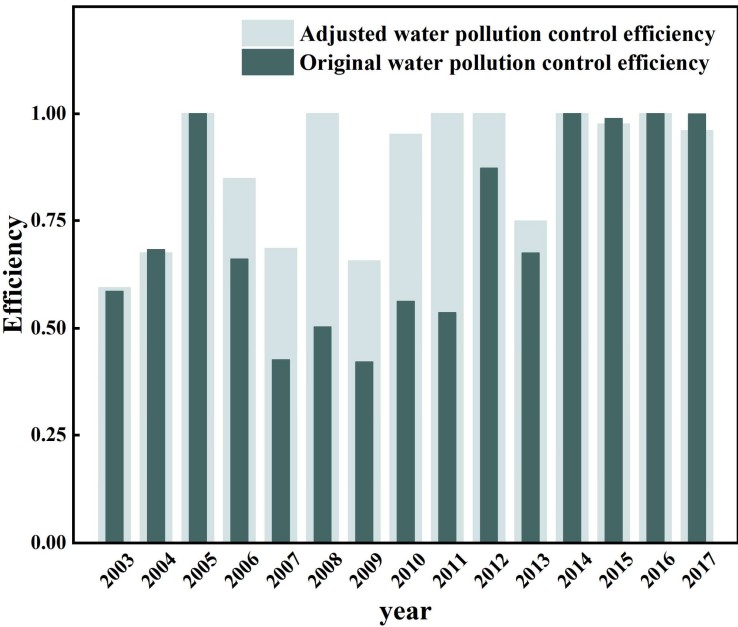

**Figure 8.** Comparison of the original and adjusted efficiency of water pollution control.

In summary, the efficiency of water pollution control appeared to increase overall after the adjustment of input indicators. By eliminating the interference of the external environment and random errors, it enabled most of the years with low efficiency in the first stage to rise in water pollution prevention efficiency, and the variation of efficiency level among the decision units decreased. It was evident that using the three-stage SBM-DEA

model to measure water pollution prevention efficiency can make the evaluation results more accurate and reliable.

*4.5. Policy Suggestions to Improve the Efficiency of Water Pollution Control*

According to the results above, the following policy recommendations can be given to further improve the efficiency of water pollution control in Chengde city. Firstly, the government should improve the efficiency of the use of funds for water pollution prevention and control. It is essential to effectively regulate and supervise the water pollution prevention funds, stipulate that the funds are earmarked for specific purposes, and clarify the rights and obligations of relevant persons. Furthermore, the improvement of the technologies for water pollution prevention and control can help break the barrier that makes it difficult to further improve the water quality, which is conducive to the enhancement of the efficiency of utilization of funds in water pollution prevention and control. The research and development of water pollution control technology and equipment suitable for local use should be emphasized, and the related senior technical staff should be trained.

Secondly, it is necessary to optimize the industrial structure and accelerate the pace of economic transformation and upgrading. The secondary industry accounted for the most significant inverse impact on the efficiency of water pollution control, and the optimization and upgrading of the industrial structure is quite urgent. The development of clean production and green production enables the industrial structure to gradually transform from a high pollution and consumption industry to an environmentally friendly one. In a word, while ensuring the continuous growth of the gross economic output, the discharge of water pollutants should be reduced as much as possible so as to improve the efficiency of pollution control.

In addition, public participation in water environmental protection should be increased. On one hand, through various forms of water environment education and practical activities, the increasing public awareness of environmental protection will force the government to carry out water pollution prevention and control more efficiently. On the other hand, the public should be encouraged to participate in environmental management to develop the power of public supervision due to the fact that the multiple participation contributes to the effective implementation of water pollution prevention and control.

## 5. Conclusions

Based on the output indicator of comprehensive water quality index, this paper measured the efficiency of water pollution control in Chengde city from 2003 to 2017 using a three-stage SBM-DEA model and analyzed the slack variables and the external environmental factors affecting the efficiency of water pollution control, and the main conclusions were as follows:

- The comprehensive water quality index of Chengde city showed a fluctuating upward trend and maintained at a high level in recent years, indicating that the water pollution prevention and control efforts in Chengde city achieved great effectiveness. Without removing external environmental variables and random factors, the efficiency of water pollution control in Chengde city also showed a general fluctuating upward trend between 2003 and 2017. The excessive input of capital and the shortfall of output of the comprehensive water quality index were the main reasons why most of the decision units failed to achieve DEA effectiveness.
- External environment and random error had significant effects on efficiency of water pollution control. The coefficients of the four external environmental variables of GDP per capita, the share of secondary industry, the share of tertiary industry, and the degree of fiscal decentralization on the slack variables of capital and labor inputs were positive, indicating that the increase in the values of these four environmental variables had a negative impact on the efficiency of water pollution control. By comparing the magnitude of the absolute values of the coefficients, it could be seen that the environmental variables with the greatest impact on capital and labor inputs

were the share of the secondary industry and the degree of fiscal decentralization, respectively.

- By comparing the input indicators of the first stage with the third stage, it could be found that the adjusted inputs were almost always greater than the initial values, which further verified the influence of external environmental variables and random errors on the efficiency of water pollution control. By comparing the efficiency of water pollution control in two different stages, it could be seen that the efficiency in stage one was significantly underestimated. By eliminating the effects of external environment and random errors, the efficiency in stage three could reflect the actual efficiency level of water pollution control in Chengde city.

**Author Contributions:** Y.C. conceived and wrote the paper; C.L. and X.Z. reviewed and revised the paper; X.L. analyzed the data; X.Z. performed the research; Q.T. reviewed and revised the paper. All authors have read and agreed to the published version of the manuscript.

**Funding:** This study is supported by the National Natural Science Foundation of China (52125902), the Program for Guangdong Introducing Innovative and Entrepreneurial Teams (2019ZT08L213) and GEF Mainstreaming Integrated Water and Environment Management (P145897-3-03).

**Institutional Review Board Statement:** Not applicable.

**Informed Consent Statement:** Not applicable.

**Data Availability Statement:** Restrictions apply to the availability of these data. Data are available from the authors with the permission of third party.

**Acknowledgments:** We appreciate all the financial support for this study. We would like to extend special thanks to the editor and reviewers for insightful advice and comments on the manuscript.

**Conflicts of Interest:** The authors declare no conflict of interest.

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
