# Peer review of "Efficiency of Water Pollution Control Based on a Three-Stage SBM-DEA Model"

_water, doi:10.3390/w14091453_

Round 1

Reviewer 1 Report

Dear authors,

The manuscript is an interesting study in the context of sustainable development, as it is necessary to ensure water for the entire population and to ensure good environmental quality. Worldwide, water pollution is one of the major environmental problems. The efficiency of water pollution control is mandatory since large investments in technologies and facilities have been increased in the last years.

I have some minor comments:

  •  There are some small corrections to be made: Line 94 (‘is measured d by”, please delete d before word by), missing point after the phrase -line 108;
  •  I suggest using researchers instead of scholars (lines 95 and 236).
  •  Line 379, please correct (*,** and **) with (*,** and ***);
  • There are some repetitions, i.e., lines 400, 414, and 428 (the calculation results showed…..), lines 422- 423 (is conducive to….. ). Please rephrase some of these paragraphs;

Thus, my decision is a MINOR REVISION.

Reviewer 2 Report

In this study, a three-stage SBM-DEA model was applied for determination of water pollution control efficiency of Chengde city, China. The reviewer has listed out several comments to include to improve the overall quality of the manuscript.

Comments:

Line 18: Abstract: Expand the acronym, SBM-DEA.

Line 44 – 45: Could you provide more background information about stochastic frontier analysis (SFA) and data envelopment analysis (DEA). Why you specifically interested in SBM-DEA model?

Line 120: “of local water pollution control activities from 2003 to 2017.” Why you have considered 2003 – 2017 time period?

Figure 1: Any reference for the reported analytical framework? Or you have developed/proposed this framework?

Line 171: Expand the acronym, OLS.

Line 244: Give full form of the acronym, GDP

Line 238: “2.3. Environmental impact factors”. Why you choose these three factors, (1) Economic development level, (2) Industrial structure, and (3) Government influence.

Line 293: Could give more description about water quality index.

The discussion of the reported results is week which needs significant improvement.

What is key difference between secondary (line 400) and tertiary (line 414) industry.

Conclusions section is quite lengthy. Revise it by providing the most important findings from this study. Give the policy recommendation as a new section (maybe before Conclusions).

Round 2

Reviewer 2 Report

No more comments.